# Risk reduction in SARS-CoV-2 infection and reinfection conferred by humoral antibody levels among essential workers during Omicron predominance

James Hollister[1]*, Cynthia Porter[1], Ryan Sprissler[2], Shawn C. Beitel[1], James K. Romine[1], Jennifer L. Uhrlaub[3], Lauren Grant[4], Young M. Yoo[4], Ashley Fowlkes[4], Amadea Britton[5], Lauren E. W. Olsho[6], Gabriella Newes-Adeyi[6], Sammantha Fuller[6], Pearl Q. Zheng[6], Manjusha Gaglani[7,8], Spencer Rose[7], Kayan Dunnigan[7], Allison L. Naleway[9], Lisa Gwynn[10], Alberto Caban-Martinez[10], Natasha Schaefer Solle[10], Harmony L. Tyner[11], Andrew L. Philips[12], Kurt T. Hegmann[12], Sarang Yoon[12], Karen Lutrick[13], Jefferey L. Burgess[1], Katherine D. Ellingson[1]

1 Mel and Enid Zuckerman College of Public Health, University of Arizona, Tucson, Arizona, United States of America, 2 University of Arizona Genetics Core–Center for Applied Genetics and Genomic Medicine, University of Arizona, Tucson, Arizona, United States of America, 3 Immunobiology, College of Medicine–Tucson, University of Arizona Health Sciences, University of Arizona, Tucson, Arizona, United States of America, 4 Influenza Division, National Center for Immunization and Respiratory Diseases, Centers for Disease Control and Prevention, Atlanta, Georgia, United States of America, 5 Coronavirus and Other Respiratory Viruses Division, National Center for Immunization and Respiratory Diseases, Centers for Disease Control and Prevention, Atlanta, Georgia, United States of America, 6 Abt Associates, Rockville, Maryland, United States of America, 7 Baylor Scott & White Health, Temple, Texas, United States of America, 8 Texas A&M University College of Medicine, Temple, Texas, United States of America, 9 Kaiser Permanente Center for Health Research, Portland, Oregon, United States of America, 10 Leonard M. Miller School of Medicine, University of Miami, Miami, Florida, United States of America, 11 St. Luke's Regional Health Care System, Duluth, Minnesota, United States of America, 12 Rocky Mountain Center for Occupational and Environmental Health, Department of Family and Preventive Medicine, University of Utah Health, Salt Lake City, Utah, United States of America, 13 Family and Community Medicine, College of Medicine–Tucson, University of Arizona Health Sciences, Tucson, Arizona, United States of America

* jameshollister@arizona.edu

**Data Availability Statement:** Data cannot be shared publicly by the authors because it is owned by the Centers for Disease Control and Prevention

## Abstract

The extent to which semi-quantitative antibody levels confer protection against SARS-CoV-2 infection in populations with heterogenous immune histories is unclear. Two nested case-control studies were designed within the multisite HEROES/RECOVER prospective cohort of frontline workers to study the relationship between antibody levels and protection against first-time post-vaccination infection and reinfection with SARS-CoV-2 from December 2021 to January 2023. All participants submitted weekly nasal swabs for rRT-PCR testing and blood samples quarterly and following infection or vaccination. Cases of first-time post-vaccination infection following a third dose of monovalent (origin strain WA-1) mRNA vaccine (n = 613) and reinfection (n = 350) were 1:1 matched to controls based on timing of blood draw and other potential confounders. Conditional logistic regression models were fit to estimate infection risk reductions associated with 3-fold increases in end titers for receptor binding domain (RBD). In first-time post-vaccination and reinfection study samples, most were female (67%, 57%), non-Hispanic (82%, 68%), and without chronic conditions (65%, 65%).

(CDC), and the data contains personal identifying information. Data are available upon request pending approval from the CDC for researchers who meet the criteria for access to confidential data. Requests for de-identified data can be sent to LTO7@cdc.gov.

**Funding:** Author JLB received funding from the US Centers for Disease Control and Prevention (award number: 75D30120C08379; URL: www.cdc.gov). Abt Associates received funding from the US Centers for Disease Control and Prevention (award number: 75D30120C08150). The US Centers for Disease Control and Prevention made key decisions on the study design with scientific and operational input from study investigators.

**Competing interests:** We have read the journal's policy and the authors of this manuscript have the following competing interests: RS reports a relationship with American Council of Life Insurers that includes: speaking and lecture fees. RS reports a relationship with California legal case Ebers v. Castle Park that includes: consulting or advisory. RS reports a relationship with Geneticure, Inc. that includes: equity or stocks. RS reports a relationship with Beckman Coulter that includes: speaking and lecture fees. RS reports a relationship with Shay Emma Hammer Research Foundation that includes: board membership. RS has patent issued to Arizona Board of Regents on Behalf of the University of Arizona. MG reports a relationship with Infectious Diseases and Immunization Committee, Texas Pediatric Society, Texas Chapter of the American Academy of Pediatrics that includes: board membership. This does not alter our adherence to PLOS ONE policies on sharing data and materials

The odds of first-time post-vaccination infection were reduced by 21% (aOR = 0.79, 95% CI = [0.66–0.96]) for each 3-fold increase in RBD end titers. The odds of reinfection associated with a 3-fold increase in RBD end titers were reduced by 23% (aOR = 0.77, 95% CI = [0.65–0.92] for unvaccinated individuals and 58% (aOR = 0.42, 95% CI = [0.22–0.84]) for individuals with three mRNA vaccine doses following their first infection. Frontline workers with higher antibody levels following a third dose of mRNA COVID-19 vaccine were at reduced risk of SARS-CoV-2 during Omicron predominance. Among those with previous infections, the point estimates of risk reduction associated with antibody levels was greater for those with three vaccine doses compared to those who were unvaccinated.

## Introduction

The Coronavirus Disease 2019 (COVID-19) pandemic has transitioned from a phase defined by acute morbidity and mortality in immunologically naïve populations to one defined by post-vaccination infections and reinfections in populations with various combinations of infection- and vaccine-induced immunity. Emergence of highly transmissible Omicron sub-variants of severe acute respiratory syndrome coronavirus 2 (SARS-CoV-2) accelerated transmission despite high levels of population immunity to the prior strains and subvariants [1]. The first vaccine trials established that higher post-vaccine antibody levels were correlated with increased protection against SARS-CoV-2 infection [2]. Other cohort studies have demonstrated variable response to the vaccines based on individual history of infection [3–5]. It is critical to understand the relationship between antibody levels and protection against Omicron post-vaccination infections and reinfections in populations with heterogenous infection and vaccination histories [6].

Conceptually, establishing and validating specific antibody levels that correlate with protection against SARS-CoV-2 infection could result in biomarkers that are reliably predictive of vaccine effectiveness [7]. For influenza, hemagglutination inhibition (HI) assay titers have proven useful in comparing different vaccines, with HI end titers of 1:40 generally accepted to be associated with a 50% reduction in risk of influenza [8, 9]. However, establishing SARS-CoV-2 antibody correlates of protection is complicated by the rapid emergence of variants and subvariants, differences in assays used to quantify antibody levels, and differential waning of antibody levels over time relative to immune conferring events [3, 10]. Nonetheless, calls for SARS-CoV-2 correlates of protection persist as questions remain regarding the durability of immunity over time given the evolution of subvariants capable of evading acquired immune responses [11, 12].

Uncertainties also remain in determining optimal timing for future COVID-19 vaccine doses [13]. This report examined the relationship between humoral antibody levels and the risks of 1) first-time post-vaccination infection following a third dose of an origin strain WA-1 monovalent mRNA vaccine; and 2) reinfection among individuals with one previous infection stratified by vaccination status at the time of reinfection (i.e., unvaccinated or had received two to three doses of mRNA vaccine) during early Omicron predominance.

## Materials and methods

### Study design

Two nested case-control studies were designed within a prospective cohort of frontline workers from eight locations in the United States (US) to study the relationship between antibody

levels and SARS-CoV-2 infections during early Omicron predominance (defined here as December 2021 through September 2022). The first study defined cases as those with a first-time SARS-CoV-2 post-vaccination infection following a third dose of COVID-19 origin strain WA-1 monovalent mRNA vaccine. The second defined cases as those infected with SARS-CoV-2 after one previous infection; at the time of reinfection, cases were either unvaccinated or had received two to three doses of COVID-19 origin strain WA-1 monovalent mRNA vaccine.

## Participants

Beginning on July 27, 2020, frontline workers were followed in prospective cohorts through the Arizona Healthcare, Emergency Response, and Other Essential workers Study (HEROES) and the Research on the Epidemiology of SARS-CoV-2 in Essential Response Personnel (RECOVER) sites in Arizona, Florida, Minnesota, Oregon, Texas, and Utah [14, 15]. Briefly, eligible participants included adults who worked at least 20 hours per week in occupations that required frequent direct contact with non-household members (i.e., healthcare workers, first responders, and other essential workers). Recruitment ended on April 15, 2023. Upon enrollment, participants completed a survey to collect baseline information about characteristics related to sociodemographic, occupation, health status, health-related behaviors, and prior SARS-CoV-2 infection. COVID-19 vaccination information was self-reported through surveys and validated through state vaccine registries or medical records. Additional surveys were completed upon infection, and self-reported information on mask use and exposures was collected monthly.

Each week and at onset of symptoms, participants provided a mid-turbinate nasal specimen that was tested at the Marshfield Clinic Laboratory (Marshfield, WI) via real time reverse transcription-polymerase chain reaction (rRT-PCR) for the presence of SARS-CoV-2. Additionally, participants were asked to submit blood samples at three frequencies: (1) upon enrollment, (2) quarterly, and (3) 28 days after any immunity-conferring event (SARS CoV-2 infection or COVID-19 vaccination).

The study protocol and procedures were reviewed and approved by the Arizona Department of Health, the University of Arizona Institutional Review Boards (IRBs), Baylor Scott and White Research Institute IRB, Kaiser Permanente Northwest IRB (IRB of record for the CDC and Abt Associates), St. Luke's Hospital Duluth IRB, University of Miami Human Subjects Research Office, and University of Utah IRB. All study participants provided informed written consent for all study activities.

## Study inclusion criteria and case definition

For the first analytic study sample, cases of first-time post-vaccination infections were defined as individuals with an incident SARS-CoV-2 infection occurring at least 14 days after a third dose origin strain WA-1 monovalent Pfizer-BioNTech COVID-19 mRNA vaccine, Moderna COVID-19 mRNA-1273 or a combination of the two vaccines among individuals with no history of previous infection. Infection history prior to enrollment was ascertained via self-report upon study enrollment and confirmed with a qualitative result from the baseline blood draw, if available, among only those unvaccinated at the time of enrollment by examining S2 and RBD (S3 and S4 Figs). If a participant was vaccinated at the time of their baseline blood draw, then we solely relied on self-reported data for pre-study infection history. Although nucleocapsid (N) titers were available for all participants with a baseline blood collection, they were unable to be used to detect prior SARS-CoV-2 infections due to low specificity and sensitivity. Variant of infection was confirmed by whole-genome sequencing for eligible specimens, or

estimated by using the state-specific predominant variant at the time of infection according to Centers for Disease Control and Prevention data [16]. In our final analytic set, approximately 50% of samples were sequenced. To be included in this study, participants had to have submitted a nasal specimen at least 80% of weeks and collected blood draws per the study protocol during the study period with no evidence of SARS-CoV-2 infection at the time of the blood draw following their third dose of vaccine. Cases were defined as participants who tested positive for SARS-CoV-2 after the blood draw and prior to any additional vaccine dose (Fig 1). Third vaccine doses occurred from June 2021 to April 2022, blood draws occurred from June 2021 to August 2022, and Omicron infections occurred from December 2021 and September 2022.

Within the second analytic study sample, reinfection cases were defined as one of three groups (1) participants who had a previous infection but were unvaccinated, (2); participants with a previous infection followed by completion of two doses of a monovalent COVID-19 mRNA vaccine (primary series only); or (3) participants with a previous infection followed by three doses of a monovalent COVID-19 mRNA vaccine (primary series and a third monovalent dose). Reinfections were defined as Omicron infections occurring at least 90 days after a previous Omicron SARS-CoV-2 infection or at least 45 days after a previous infection of a distinct variant (e.g., WA-1 vs. Delta vs. Omicron). Variant of first infection and reinfection was confirmed by either whole-genome sequencing for eligible specimens, or estimated by using the state-specific predominant variant at the time of infection according to Centers for Disease Control and Prevention data [16]. Those receiving a Janssen Ad26.COV2.S COVID-19 vaccine and those with only one mRNA vaccination were ineligible, thus included cases had none, two, or three immunizations. Participants were eligible for the reinfection study sample set if they: (1) had either reported testing positive for SARS-CoV-2 prior to enrollment or tested positive by rRT-PCR during the study period prior to COVID-19 vaccination, and (2) had a blood draw at any time after their last immune conferring event (first infection for unvaccinated participants or last vaccine dose for vaccinated participants). For this group, a case was defined as a participant with a confirmed reinfection after the blood draw (Fig 2). Initial infections occurred from March 2020 to August 2022, blood draws occurred from August 2020 to November 2022, and reinfections occurred from December 2021 to January 2023.

## Matching

The final analytic samples were created by matching eligible cases to controls in a 1:1 ratio. Matched controls for the first-time post-vaccination cases included individuals with no history of infection who were matched by study site and days between the third origin strain WA-1 monovalent COVID-19 mRNA vaccine and blood draw. Matched controls for reinfection cases included individuals with a history of one infection who were matched to cases by: study site, variant of first infection (origin strain WA-1, Delta, or Omicron), vaccination status at the time of blood draw, and days between last immune conferring event (first infection for unvaccinated participants or last A vaccine dose for vaccinated participants) and blood draw. For both groups, time windows were used to match days between immune conferring events and blood draws rather than exact matches. For blood samples that were collected less than 150 days since the last immune conferring events, time difference windows of 3, 7, 14, and 21 days were each attempted to find the highest match quality. For blood samples collected at least 150 days since the last immune conferring events, no strict time differences were used.

## Semi-quantitative serological measures

All sera from all sites were sent to the University of Arizona Genetics Core laboratory for testing using a locally-developed and validated semi-quantitative enzyme-linked immunosorbent

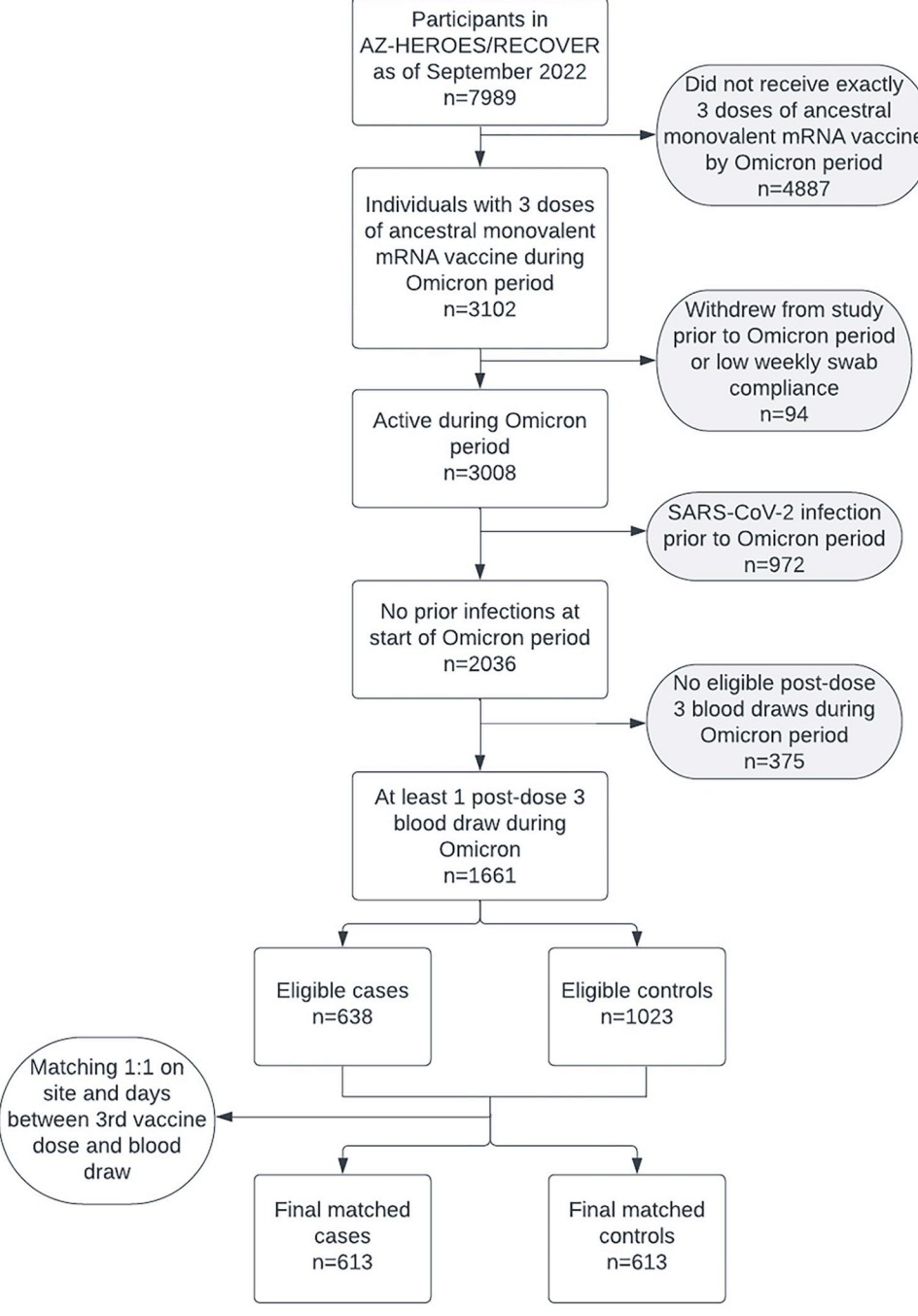

**Fig 1. Flowcharts showing inclusion and exclusion criteria for nested case-control studies within the HEROES/ RECOVER cohort of frontline workers: 613 cases of first-time post-vaccination infection following a third dose of COVID-19 origin strain WA-1 monovalent mRNA vaccine were matched 1:1 to controls based on site and time between vaccination and blood draw.**

assay (ELISA) to measure antibody binding to the receptor binding domain (RBD) and S2 sub-unit domain (S2) of the SARS-CoV-2 Washington-1 spike protein, as previously described [17]. Five 3-fold dilutions of immune sera were made starting at a 1:60 dilution and ending at 1:4860. End titer values represent the first dilution in which an immune response was no longer detected. Those with immune response still detected at the final dilution were imputed to

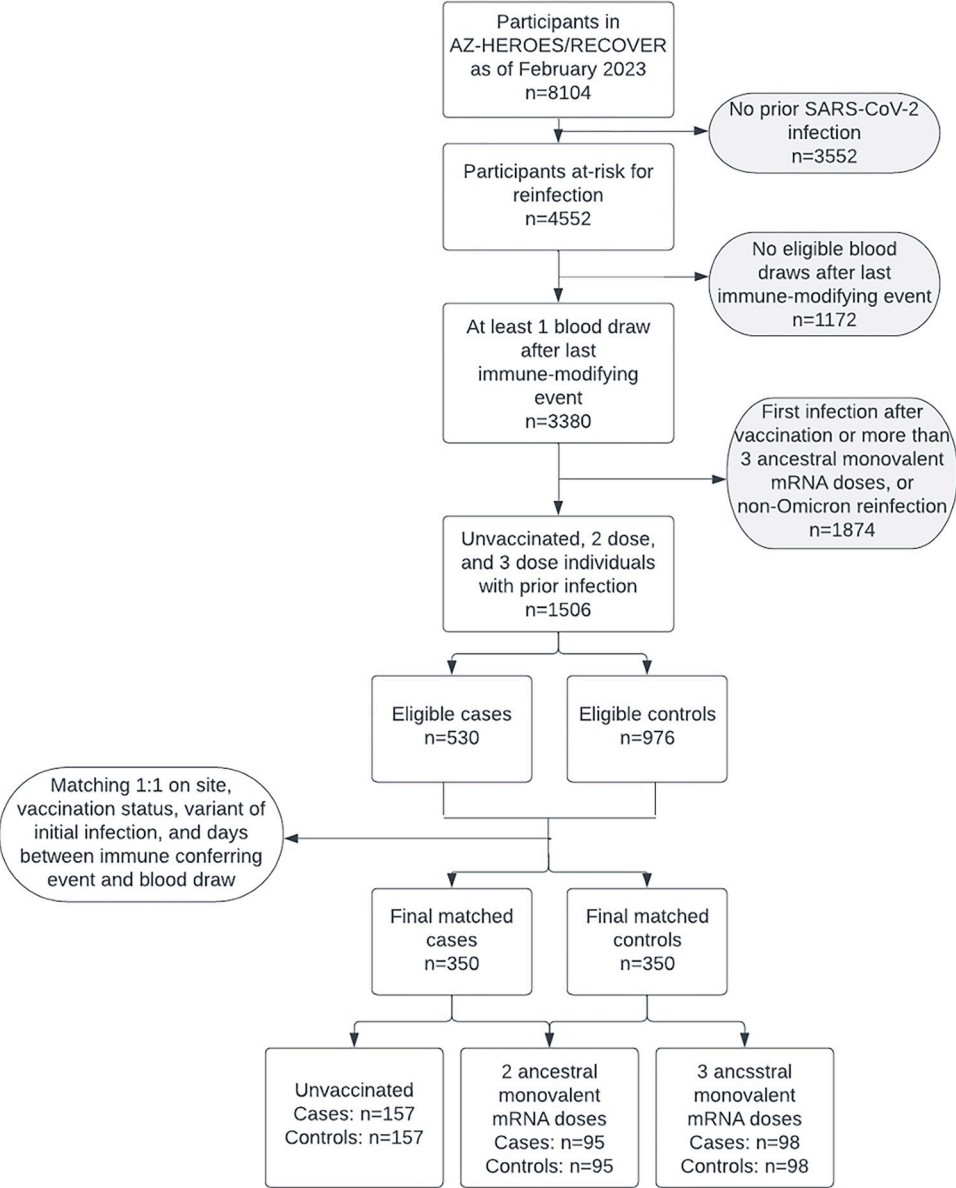

**Fig 2. Flowcharts showing inclusion and exclusion criteria for nested case-control studies within the HEROES/ RECOVER cohort of frontline workers: 350 cases of reinfection were matched 1:1 to controls based on site, vaccination status, variant of first infection, and time between immune conferring event (either vaccine or infection) and blood draw.**

1:9720. Additionally, RBD and S2 antibody levels were measured as the area under the serial dilution curve (AUC). This continuous measurement represents a weighted sum of the optical density value at each dilution. Both end titer and AUC measurements of RBD and S2 were included in the present analysis.

## Statistical analysis

Analyses were conducted using SAS Studio v9.4 (SAS Institute Inc., Cary, North Carolina USA), GraphPad Prism v9.5.1 (GraphPad Software, Boston, Massachusetts USA), and R v4.2.0 (R Core Team, Vienna, Austria). Demographic information and measured antibody response

for cases and controls was compared using paired t tests and univariate conditional logistic regression for continuous and categorical variables. The outcomes of first-time post-vaccination infection and reinfection with Omicron were modeled using conditional logistic regression to assess the unadjusted and adjusted effects of RBD and S2 humoral antibody levels. Adjusted odds ratios were calculated with models including the following variables that were selected a priori as putative confounders: gender, age (18–49 or over 50 years), presence of at least one chronic condition (i.e. asthma, chronic lung disease, cancer, diabetes, heart disease, hypertension, immunosuppression, kidney disease, liver disease, neurologic or neuromuscular disease or disorder, and autoimmune disease), high exposure to individuals infected with SARS-CoV-2 (above the cohort mean number of weekly hours), high reported mask use in the community (above the cohort mean percentage), and high reported personal protective equipment (PPE) adherence at work (above the cohort mean percentage), where reported PPE adherence at work was defined as the percentage of time in which an individual uses the PPE recommended by their employer when in direct contact with people. For the reinfection study sample, separate models were fit for each category of vaccination status: unvaccinated, 2 doses, and 3 doses of the origin strain WA-1 monovalent COVID-19 mRNA vaccine. In addition, we conducted a likelihood ratio test to assess the interaction term between vaccination status and antibody titer within the reinfection sample.

AUC antibody values were standardized by z-scores prior to modeling for ease of interpretation. For end titer data, a log base 3 transformation was used due to the three-fold dilution assay and was treated as a continuous value. Curves displaying the estimated percent reduction in risk were estimated for both first-time post-vaccination infection and reinfection models using an end titer level of 1:60 as the reference. The estimated percent reduction in risk was calculated at each titer level. The 95% confidence interval bounds on the curves were estimated when percent reduction in risk at each titer level was statistically significant at the 0.05 alpha level [2, 18]. Using risk reduction curves, we estimated the titer level for a 50% reduction in risk and reported the titer levels used in the ELISA in which the estimate fell between, as is standard in a typical correlates of protection analysis [8, 18, 19].

## Results

### First-time post-vaccination infection with Omicron following third dose of COVID-19 origin strain WA-1 monovalent mRNA vaccine

The incidence of first-time vaccine post-vaccination infections with the Omicron variant following a third dose of origin strain WA-1 monovalent COVID-19 mRNA vaccine was 1.68 per 1000 person-days (830 infections) in the HEROES/RECOVER cohort. In the sample set of 613 first-time post-vaccination cases and 613 matched controls, most participants were female (67%), white and non-Hispanic (82%), healthcare workers (64%), and had no chronic conditions (65%) (Table 1). There was a small but statistically significant difference in days between third vaccination and blood draw (41.4 days for cases and 42.4 days for controls, p = 0.003). Compared to cases, controls were slightly older (46y vs. 43y, p<0.001), and had modestly higher reported mask use in the community (61.1% vs. 57.2%) and higher reported PPE adherence at work (81.6% v. 77.0%) (Table 1).

For cases, the mean number of days between the third vaccine dose and infection was 183 days. Compared to cases, a higher percentage of controls had the maximum end titer values (1:9720) for RBD (65.1% vs. 56.9%,) and S2 (41.1% vs. 32.0%), but the distribution between cases and controls for RBD end titer values was not statistically significant. Controls had significantly higher mean RBD (0.0124 vs. 0.0118) and S2 (0.0102 vs. 0.0097) AUC values compared to cases (Table 2, S1 Fig).

**Table 1. Demographic and health characteristics for nested case-control samples within the HEROES/RECOVER prospective cohort of frontline workers: the first-time post-vaccination sample was matched by site and days between booster vaccine and blood draw, and the reinfection sample was matched on site, vaccination status, variant of first infection, and days between most recent immune conferring event and blood draw.**

| | Sample | | | | | |
| --- | --- | --- | --- | --- | --- | --- |
| | First-time post-vaccination infection | | | Reinfection | | |
| | Cases (n = 613) | Controls (n = 613) | p-value[a] | Cases (n = 350) | Controls (n = 350) | p-value[a] |
| **Site, n (%)** | | | 1.00 | | | 1.00 |
| Tucson, AZ | 145 (23.7) | 145 (23.7) | | 123 (35.1) | 123 (35.1) | |
| Phoenix, AZ | 73 (11.9) | 73 (11.9) | | 46 (13.1) | 46 (13.1) | |
| Other, AZ | 37 (6.0) | 37 (6.0) | | 28 (8.0) | 28 (8.0) | |
| Florida | 22 (3.6) | 22 (3.6) | | 50 (14.3) | 50 (14.3) | |
| Minnesota | 132 (21.5) | 132 (21.5) | | 30 (8.6) | 30 (8.6) | |
| Oregon | 37 (6.0) | 37 (6.0) | | 5 (1.4) | 5 (1.4) | |
| Texas | 33 (5.4) | 33 (5.4) | | 22 (6.3) | 22 (6.3) | |
| Utah | 134 (21.9) | 134 (21.9) | | 46 (13.1) | 46 (13.1) | |
| **Vaccination status[b]** | | | 1.00 | | | 1.00 |
| Unvaccinated | 0 (0.0) | 0 (0.0) | | 157 (44.9) | 157 (44.9) | |
| 2 Doses | 0 (0.0) | 0 (0.0) | | 95 (27.1) | 95 (27.1) | |
| 3 Doses | 613 (100.0) | 613 (100.0) | | 98 (28.0) | 98 (28.0) | |
| **Variant of first infection** | | | NA | | | 1.00 |
| Origin strain WA-1 | NA | NA | | 267 (76.3) | 267 (76.3) | |
| Delta | NA | NA | | 48 (13.7) | 48 (13.7) | |
| Omicron | 613 (100) | NA | | 35 (10.0) | 35 (10.0) | |
| **Days from ICE to blood draw, mean (SD)[c]** | 41.4 (39.8) | 42.4 (40.6) | 0.0003 | 113.2 (108.9) | 129.3 (136.3) | <0.001 |
| **Age (y), mean (SD)** | 43.0 (10.0) | 46.4 (11.9) | <0.001 | 45.6 (11.1) | 44.3 (11.6) | 0.10 |
| **Gender, n (%)** | | | 0.60 | | | 0.16 |
| Female | 415 (67.7) | 406 (66.2) | | 189 (54.0) | 208 (59.4) | |
| Male | 197 (32.1) | 205 (33.4) | | 159 (45.4) | 142 (40.6) | |
| Other | 1 (0.2) | 2 (0.3) | | 0 (0.0) | 0 (0.0) | |
| Missing | 0 (0.0) | 0 (0.0) | | 2 (0.6) | 0 (0.0) | |
| **Race/ethnicity, n (%)** | | | 0.24 | | | 0.41 |
| Non-Hispanic, White | 489 (79.8) | 512 (83.5) | | 232 (66.3) | 245 (70.0) | |
| Hispanic | 72 (11.7) | 52 (8.5) | | 89 (25.4) | 72 (20.6) | |
| Non-Hispanic, Black | 18 (2.9) | 19 (3.1) | | 11 (3.1) | 10 (2.9) | |
| Non-Hispanic, Asian | 18 (2.9) | 23 (3.8) | | 3 (0.9) | 7 (2.0) | |
| Other | 8 (1.3) | 5 (0.8) | | 4 (1.1) | 5 (1.4) | |
| Missing | 8 (1.3) | 2 (0.3) | | 11 (3.1) | 11 (3.1) | |
| **Occupation, n (%)[d]** | | | 0.06 | | | 0.06 |
| Healthcare Worker | 376 (61.3) | 411 (67.0) | | 125 (35.7) | 145 (41.4) | |
| First Responder | 79 (12.9) | 60 (9.8) | | 117 (33.4) | 92 (26.3) | |
| Other Essential Worker | 158 (25.8) | 142 (23.2) | | 108 (30.9) | 113 (32.3) | |
| **Chronic conditions, n (%)[e]** | | | 0.91 | | | 0.24 |
| None | 403 (65.7) | 397 (64.8) | | 237 (67.7) | 216 (61.7) | |
| One | 114 (18.6) | 113 (18.4) | | 73 (20.9) | 74 (21.1) | |
| Two Or More | 93 (15.2) | 98 (16.0) | | 33 (9.4) | 44 (12.6) | |
| Missing | 3 (0.5) | 5 (0.8) | | 7 (2.0) | 16 (4.6) | |
| **Immunosuppressive medication, n (%)[f]** | | | 0.70 | | | 0.82 |
| Yes | 14 (2.3) | 12 (2.0) | | 11 (3.1) | 10 (2.9) | |
| No | 590 (96.2) | 592 (96.6) | | 330 (94.3) | 320 (91.4) | |

*(Continued)*

**Table 1.** (Continued)

| | Sample | | | | | |
|---|---|---|---|---|---|---|
| | First-time post-vaccination infection | | | Reinfection | | |
| | Cases (n = 613) | Controls (n = 613) | p-value[a] | Cases (n = 350) | Controls (n = 350) | p-value[a] |
| Missing | 9 (1.5) | 9 (1.5) | | 9 (2.6) | 20 (5.7) | |
| **Days from ICE to infection, mean (SD)[e]** | 183.0 (80.6) | NA | NA | 341.0 (180.6) | NA | NA |
| **Avg weekly hrs. exposed to COVID, mean (SD)[g]** | 4.8 (9.9) | 5.0 (10.4) | 0.74 | 7.0 (12.0) | 6.2 (10.5) | 0.51 |
| **Avg % adherence to PPE rules at work, mean (SD)[h]** | 77.0 (26.4) | 81.6 (24.3) | 0.001 | 58.2 (31.7) | 62.3 (32.1) | 0.09 |
| **Avg % time masked in public, mean (SD)** | 57.2 (29.1) | 61.1 (29.7) | 0.01 | 41.7 (28.7) | 47.3 (32.6) | 0.02 |

[a]From paired t-test and conditional logistic regression for continuous and categorical variables, respectively.

[b]All vaccine doses are monovalent origin strain WA-1 mRNA vaccines.

[c]ICE (immune conferring event) is the third vaccine dose for the first-time post-vaccination infection with Omicron case-control cohort and the 3 dose strata within the reinfection case-control cohort, the second vaccine dose for the 2 dose strata within the reinfection case-control cohort, and the initial SARS-CoV-2 infection for the unvaccinated strata within the reinfection case-control cohort.

[d]Other essential workers include occupation sectors with potentially high exposures to SARS-CoV-2 such as education, agriculture, public transportation services, waste collection, delivery, utilities, community-based services, childcare, and others

[e]Chronic conditions include asthma, chronic lung disease, cancer, diabetes, heart disease, hypertension, immunosuppression, kidney disease, liver disease, neurologic or neuromuscular disease or disorder, and autoimmune disease.

[f]Participants are asked "Are you currently taking prednisone or other ongoing steroid medications (excluding inhaled steroids and one-time injections) or any other medications that may suppress your body's ability to fight infection?".

[g]Exposure to individuals infected with SARS-CoV-2.

[h]Reported personal protective equipment (PPE) adherence at work was defined as the percentage of time in which an individual uses the PPE recommended by their employer when in direct contact with people

Unadjusted and adjusted conditional logistic regression models for first-time vaccine-post-vaccination infection with Omicron infection following a third dose of an origin strain WA-1 monovalent COVID-19 mRNA vaccine showed statistically significant protective effects for RBD and S2 end titers. Each three-fold increase in RBD end titer reduced the odds of first-time post-vaccination infection with Omicron by 21% (aOR: 0.79, 95% CI: [0.66, 0.96]). Each three-fold increase in S2 end titer reduced the odds of post-vaccination infection by 25% (aOR: 0.75, 95% CI: [0.63, 0.91]) (Table 3). Compared to the reference end titer (no antibodies detected at the 1:60 dilution), estimated risk reduction was 50% at RBD and S2 end titers between 1:540 and 1:1620 (Fig 3). Models including AUC data as our antibody measurement had similar results. Full results including estimates for all variables included in the models can be found in the supplemental material (S3 and S4 Tables).

## Reinfection

The sample set of 350 cases of reinfection and 350 matched controls included 157 pairs who were unvaccinated, 95 pairs who had completed the primary series of two doses of COVID-19 origin strain WA-1 monovalent mRNA vaccine, and 98 pairs who had received 3 doses. In the general HEROES/RECOVER cohort, the incidence of reinfections with the Omicron variant for individuals who were unvaccinated, had 2 doses, and had 3 doses was 0.98, 0.71, and 0.71 per 1000 person-days, respectively. Overall, most participants were female (57%), white and non-Hispanic (68%), and had no chronic conditions (65%). Most cases and controls were first infected with the origin strain WA-1 strain of SARS-CoV-2 (76%) (Table 1). Although cases and controls were matched on the interval between the most recent immune conferring event (first infection or vaccination) and the date of their blood draw, there was a nominal but

**Table 2. Quantitative antibody results, including end titers and area under the curve (AUC) calculations for SARS-CoV-2 receptor-binding domain (RBD) and (S2) spike protein subunits in nested case-control samples from the HEROES/RECOVER prospective cohort of frontline workers.**

| Variable | Sample | | | | | |
|---|---|---|---|---|---|---|
| | First-time post-vaccination | | | Reinfection | | |
| | Cases (n = 613) | Controls (n = 613) | p-value[a] | Cases (n = 350) | Controls (n = 350) | p-value[a] |
| **RBD End Titer, n (%)** | | | 0.08 | | | 0.003 |
| 1:60 | 4 (0.7) | 3 (0.5) | | 36 (10.3) | 19 (5.4) | |
| 1:180 | 5 (0.8) | 3 (0.5) | | 31 (8.9) | 25 (7.1) | |
| 1:540 | 14 (2.3) | 13 (2.1) | | 43 (12.3) | 38 (10.9) | |
| 1:1620 | 45 (7.3) | 39 (6.4) | | 55 (15.7) | 51 (14.6) | |
| 1:4860 | 196 (32.0) | 156 (25.4) | | 66 (18.9) | 77 (22.0) | |
| 1:9720[B] | 349 (56.9) | 399 (65.1) | | 119 (34.0) | 140 (40.0) | |
| **RBD AUC, mean (SD)** | 0.0118 (0.003) | 0.0124 (0.003) | <0.0001 | 0.0088 (0.005) | 0.0098 (0.005) | 0.0002 |
| **S2 End Titer, n (%)** | | | 0.002 | | | 0.003 |
| 1:60 | 7 (1.1) | 2 (0.3) | | 9 (2.6) | 2 (0.6) | |
| 1:180 | 8 (1.3) | 6 (1.0) | | 3 (0.9) | 4 (1.1) | |
| 1:540 | 24 (3.9) | 21 (3.4) | | 26 (7.4) | 25 (7.1) | |
| 1:1620 | 120 (19.6) | 90 (14.7) | | 95 (27.1) | 64 (18.3) | |
| 1:4860 | 258 (42.1) | 242 (39.5) | | 95 (27.1) | 124 (35.4) | |
| 1:9720[B] | 196 (32.0) | 252 (41.1) | | 122 (34.9) | 131 (37.4) | |
| **S2 AUC, mean (SD)** | 0.0097 (0.003) | 0.0102 (0.003) | 0.0003 | 0.0102 (0.004) | 0.0110 (0.003) | 0.0005 |

[a]From paired t-test and conditional logistic regression for continuous and categorical variables, respectively.

[b]Samples where antibody response was still detected at the final dilution were imputed to 1:9720.

statistically significant difference (113 days for cases and 129 days for controls, p<0.001) (Table 1) due to being matched on intervals as described previously in the methods. Compared to cases, controls had a higher percentage of individuals at the highest imputed end titer level (1:9720) (RBD: 40.0% vs. 34.0%; S2: 37.4% vs. 34.9%) and had higher mean AUC values for RBD (0.0098 vs. 0.0088) and S2 (0.0110 vs. 0.0102) (Table 2 and S2 Fig). Demographic information stratified by vaccination status can be found in the supplemental material (S1 and S2 Tables).

In unadjusted and adjusted conditional logistic regression models stratified by vaccination status, the strongest protective effects of humoral antibody levels against reinfection were noted among those with a third dose of vaccine; among those unvaccinated or with two doses of vaccine, protective effects of antibody levels were suggested but statistical significance was not consistent across all groups. Likelihood ratio tests assessing the interaction between vaccination status and antibody titers had p-values of 0.12 and 0.07 for RBD and S2 end titer, respectively, indicating evidence of a potential difference by vaccination status. For unvaccinated individuals, each three-fold increase in RBD and S2 end titer value was estimated to reduce the odds of reinfection by 23% and 18%, respectively, although the estimate for S2 was not statistically significant (RBD aOR: 0.77, 95% CI: [0.65, 0.92]; S2 aOR: 0.82, 95% CI: [0.64, 1.04]) (Table 3). For unvaccinated individuals, compared to an RBD end titer value of 1:60, risk was reduced by 50% between end titer values of 1:540 and 1:1620, respectively (Fig 4).

For individuals vaccinated with two doses of vaccine following their first infection, each three-fold increase in RBD and S2 end titer value was estimated to reduce the odds of reinfection by 27% and 16%, respectively; however, both of these estimates were not statistically significant (RBD aOR: 0.73, 95% CI: [0.44, 1.21]; S2 aOR: 0.84, 95% CI: [0.54, 1.30]) (Table 3).

**Table 3. Unadjusted and adjusted odds ratios (ORs) generated by conditional logistic regression models for a nested sample of individuals with first-time post-vaccination infection following third COVID-19 origin strain WA-1 monovalent mRNA vaccine dose and matched controls (n = 1226) and for a nested sample of individuals with reinfection and matched controls, stratified by vaccination status (n = 700) from the HEROES/RECOVER prospective cohort of frontline workers.**

| Study Sample | N | Outcome | Predictor | RBD | | S2 | |
|---|---|---|---|---|---|---|---|
| | | | | *Unadjusted OR (95% CI)* | *Adjusted OR (95% CI)* | *Unadjusted OR (95% CI)* | *Adjusted OR (95% CI)* |
| Never Infected, 3 doses origin strain WA-1 monovalent mRNA vaccine[a] | 1226 | Post-Vaccination Omicron Infection | End Titer | 0.82 (0.68, 0.98)* | 0.79 (0.66, 0.96)* | 0.74 (0.63, 0.87)* | 0.75 (0.63, 0.88)* |
| | | | AUC | 0.78 (0.69, 0.88)* | 0.77 (0.67, 0.88)* | 0.80 (0.70, 0.90)* | 0.79 (0.69, 0.90)* |
| Prior Infection, Unvaccinated[b] | 314 | Reinfection | End Titer | 0.77 (0.66, 0.91)* | 0.77 (0.65, 0.92)* | 0.80 (0.63, 1.01) | 0.82 (0.64, 1.04) |
| | | | AUC | 0.63 (0.46, 0.88)* | 0.64 (0.44, 0.93)* | 0.75 (0.59, 0.96)* | 0.72 (0.54, 0.94)* |
| Prior Infection + 2 doses origin strain WA-1 monovalent mRNA vaccine[c] | 190 | Reinfection | End Titer | 0.75 (0.47, 1.21) | 0.73 (0.44, 1.21) | 0.89 (0.59, 1.35) | 0.84 (0.54, 1.30) |
| | | | AUC | 0.86 (0.56, 1.32) | 0.91 (0.55, 1.50) | 0.83 (0.58, 1.19) | 0.91 (0.61, 1.34) |
| Prior Infection + 3 doses origin strain WA-1 monovalent mRNA vaccine[d] | 196 | Reinfection | End Titer | 0.39 (0.20, 0.77)* | 0.42 (0.22, 0.84)* | 0.41 (0.22, 0.79)* | 0.41 (0.21, 0.80)* |
| | | | AUC | 0.45 (0.25, 0.81)* | 0.51 (0.28, 0.94)* | 0.51 (0.32, 0.83)* | 0.56 (0.34, 0.92)* |

Abbreviations: OR: odds ratio; CI: confidence interval

Adjusted models include age, gender, presence of at least 1 chronic condition, weekly exposure to individuals infected with SARS-CoV-2, personal protective equipment (PPE) usage at work and in the community.

Odds ratio represents odds of being a case for each 3-fold increase in end titer.

[a]Cases were defined as individuals who became infected with Omicron after receiving three origin strain WA-1 monovalent COVID-19 vaccine doses and no prior infections. Cases and controls were matched on number of days between blood draw and third vaccine dose, and study site.

[b]Cases were defined as individuals who became reinfected with SARS-CoV-2 while unvaccinated. Both cases and controls were unvaccinated at the time of their blood draw.

[c]Cases were defined as individuals who became reinfected with SARS-CoV-2. Both cases and controls were unvaccinated at time of initial infection, then received 2 doses of an origin strain WA-1 monovalent COVID-19 vaccine prior to any potential reinfection. Blood draw for both cases and controls occurred after the 2nd dose.

[d]Cases were defined as individuals who became reinfected with SARS-CoV-2. Both cases and controls were unvaccinated at time of initial infection, then received 3 doses of an origin strain WA-1 monovalent COVID-19 vaccine prior to any potential reinfection. Blood draw for both cases and controls occurred after the 3rd dose.

*Statistically significant at alpha = 0.05

For individuals with three origin strain WA-1 monovalent COVID-19 mRNA vaccine doses following their first infection, each three-fold increase in RBD and S2 end titer values reduced the estimated odds of reinfection by 58% and 59%, respectively (RBD aOR: 0.42, 95% CI: [0.22, 0.84]; S2 aOR: 0.41, 95% CI: [0.21, 0.80]) (Table 3). Compared to RBD and S2 end titer values of 1:60, risk was reduced by 50% at end titer values between 1:60 and 1:180 for each (Fig 3). Models including AUC data as our antibody measurement had similar results. Full results including estimates for all variables included in the models can be found in the supplemental material (S5 and S6 Tables).

## Discussion

In this analysis of frontline workers from eight US locations, higher levels of RBD antibodies were associated with protection against first-time post-vaccination infection and reinfection with Omicron. The risk of first-time post-vaccination infection with Omicron following a third dose of monovalent mRNA vaccine was reduced by 21% for each three-fold increase in RBD end titer, with estimated 50% risk reductions achieved at end titers between 1:540 and 1:1620. The magnitude of risk reduction against reinfection depended on vaccination status, with some evidence that this difference was significant based on p-values from likelihood ratio tests of 0.12 and 0.07 for RBD and S2 end titers, respectively. For unvaccinated individuals with a prior infection, a 3-fold increase in RBD reduced reinfection risk by 23%. There were

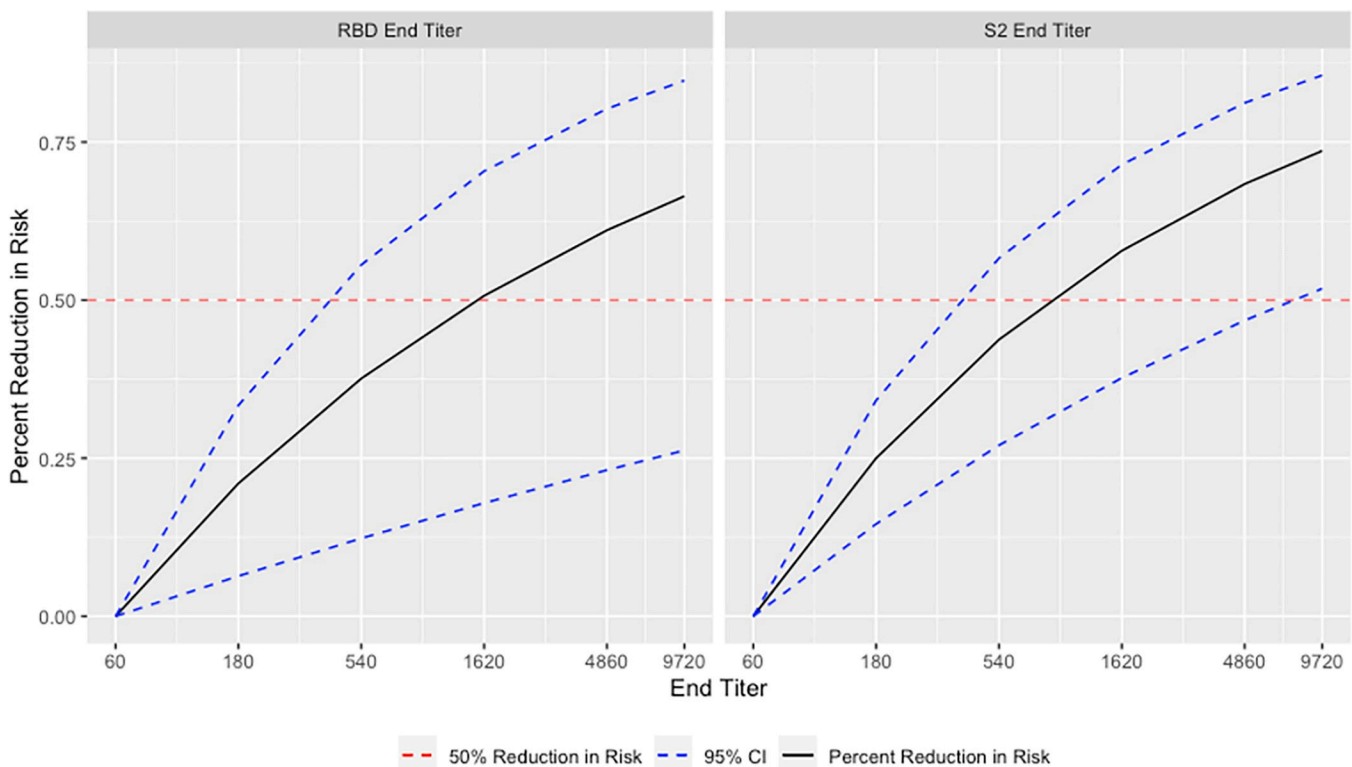

**Fig 3. Adjusted relative risk reduction for first-time post-vaccination infection with Omicron following a third dose of an origin strain WA-1 monovalent mRNA COVID-19 vaccine among frontline workers in the HEROES/RECOVER cohort with 95% confidence intervals using an end titer level of 1:60 as the reference.** Samples where antibody response was still detected at the final dilution were imputed to 1:9720. The estimated percent reduction was 50% near the 1:1620 dilution for RBD and between the 1:540 and 1:1620 dilution for S2.

no statistically significant protective effects for those with a previous infection and two origin strain WA-1 monovalent COVID-19 mRNA doses following infection. However, for individuals with three vaccine doses following their first infection, a 3-fold increase in RBD end titer value reduced reinfection risk by 58%, with estimated 50% reductions in SARS-CoV-2 reinfection risk occurring at end titers between 1:60 and 1:180. The protective effect of higher antibody levels was robust in those with a third dose of vaccine, regardless of previous infection status. However, those with previous infection showed a 50% reduction in risk at lower titers compared to those who were not previously infected, though we could not test this difference for statistical significance [4].

These findings build on previous literature reporting correlates of protection against SARS-CoV-2 infection in different populations and at different points during the pandemic. Early vaccine efficacy trials in the US found higher post-vaccination end titers in vaccine recipients to be associated with higher vaccine efficacy [2]. Similarly, early SARS-CoV-2 post-vaccination infections associated with the Alpha variant among healthcare workers in Israel who received two doses of the BNT162b2 vaccine were associated with lower neutralizing antibody titers in the peri-infection period [20]. In a household transmission study of Delta variant infections among individuals who received one or two doses of the BNT162b2 vaccine, both quantitative ELISA and neutralizing antibody titers were protective against secondary Delta infection, with titers over 1:1000 providing an immunologic threshold for protection [21]. Other studies examining the role of antibody levels and functional protection against Omicron infection among individuals who had previously received an mRNA vaccine have shown

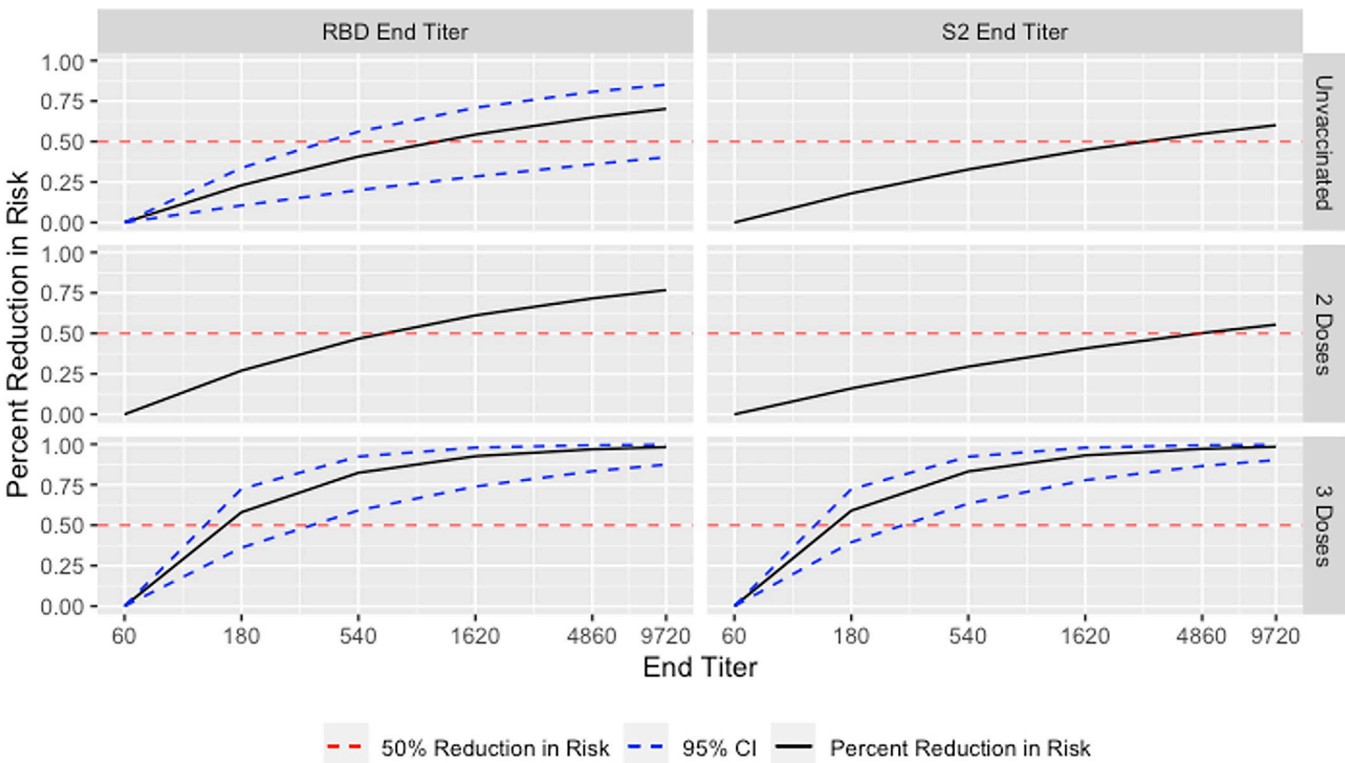

**Fig 4. Adjusted relative risk reduction in reinfection with Omicron among frontline workers in the HEROES/RECOVER cohort infection by humoral antibody titer levels with 95% confidence intervals[a] using end titer level of 1:60 as the reference and stratified by number of origin strain WA-1 monovalent mRNA COVID-19 vaccine doses.** Samples where antibody response was still detected at the final dilution were imputed to 1:9720. Among unvaccinated individuals, the estimated percent reduction was 50% between the 1:540 and 1:1620 dilution for RBD. Among individuals with 3 prior COVID-19 vaccine doses, the estimated percent reduction was 50% between the 1:60 and 1:180 dilution. [a]95% confidence interval only available for models with a statistically significant coefficient for end titer when alpha is 0.05.

mixed results [22, 23]. One potential explanation as to why serologic antibody levels could be less protective against Omicron infection is that, compared to previous variants, Omicron is relatively confined to the upper respiratory tract and therefore higher mucosal antibody threshold levels could be required to prevent infection [24]. This provides a potential explanation for why 50% risk reductions in this study were achieved at higher end titer levels for individuals with three vaccines and no previous infection compared to lower titer levels for those with a previous infection followed by three vaccine doses. These findings in combination with previous studies underscore the importance of considering both vaccination and infection history, as well as number of immune conferring events, when estimating functional protection against SARS-CoV-2 infection [25–29].

This study is subject to several limitations. First, because the semi-quantitative ELISA assessed only five dilutions, and many individuals had an immune response at the highest dilution, ability to differentiate individuals with a response above the highest dilution is limited. Additionally, treating this outcome as a continuous measure assumes linearity in the log-odds for each 3-fold increase in titer level. However, our models with AUC as the outcome, which is less sensitive to any potential ceiling effects, had similar results to the models using end titer data (S3 and S5 Tables), which helps confirm our end titer results [30]. Second, neutralizing antibodies are typically the gold standard in assessing correlates of protection, and for these analyses, a surrogate measure of semi-quantitative ELISA was used. However, antibodies measured via the semi-quantitative ELISA assay used in this study are known to correlate closely

with neutralization of live SARS-CoV-2 virus [17, 31]. We only ran ELISAs against the origin strain WA-1 strain of SARS-CoV-2, meaning that specific antibodies against the Omicron variant may differ from the results found here. Third, although the reinfection study data were matched on the interval between the most recent immune conferring event (first infection or vaccination) and blood draw, the residual differences in those times between the cases and controls may have provided some residual confounding. Additionally, differences in mean time intervals between the two analytic samples, as well as waning antibodies for samples that were further away from the previous immune-conferring event, may affect direct comparisons in results. Fourth, the HEROES/RECOVER cohort consists of frontline workers, and therefore these findings may not be generalizable beyond relatively healthy populations and populations with regular exposure to SARS-CoV-2. Due to this healthy population, we were unable to evaluate severe disease as an outcome. Finally, participants are subject to misclassification when determining eligibility for the two case-control cohorts and case or control status. Since we were limited to using RBD and S2 when determining prior infection at baseline, we solely relied on self-reported data for individuals who were vaccinated at baseline, and misclassification could occur if any of these participants incorrectly reported or failed to report a pre-study infection. Strengths of this study include its large sample size, its representativeness of Hispanic/Latino populations consistent with US demographic trends, and weekly active surveillance for infections regardless of symptoms.

In a population of frontline workers with diverse immunological histories, those with higher antibody levels had consistently lower risk of Omicron infection. Antibody correlates of protection against first-time post-vaccination infection and reinfection could be estimated for individuals who had received a third dose of a vaccine: odds of first-time post-vaccination infection were reduced by 21% for each 3-fold increase in RBD end titers, while the odds of reinfection were reduced by 58% for each 3-fold increase in RBD end titers for individuals with three mRNA vaccine doses following their first infection. Although we were able to estimate these reductions separately for each group, we were unable to statistically compare these two cohorts. Findings underscore the importance of immunologic heterogeneity, and contribution of hybrid immunity, when estimating functional protection against SARS-CoV-2 infection. Future studies could assess the Omicron-specific correlates of protection using surrogate or authentic neutralization assays, which could enable more credible interpretations about functional protection against currently circulating strains, especially with the continued updates to COVID-19 vaccines [32].

## Supporting information

**S1 Table. Demographic information for reinfection correlates of protection nested case-control cohort from AZHEROES/RECOVER, stratified by vaccination status.** [a]All vaccine doses are monovalent origin strain WA-1 mRNA vaccines. [b]Other essential workers include occupation sectors with potentially high exposures to SARS-CoV-2 such as education, agriculture, public transportation services, waste collection, delivery, utilities, community-based services, childcare, and others. [c]Chronic conditions include asthma, chronic lung disease, cancer, diabetes, heart disease, hypertension, immunosuppression, kidney disease, liver disease, neurologic or neuromuscular disease or disorder, and autoimmune disease. [d]Participants are asked "Are you currently taking prednisone or other ongoing steroid medications (excluding inhaled steroids and one-time injections) or any other medications that may suppress your body's ability to fight infection?". [e]ICE (immune conferring event) is the third vaccine dose for the first-time post-vaccination infection with Omicron cohort and the 3 dose strata within the reinfection cohort, the second vaccine dose for the 2 dose strata within the reinfection cohort,

and the initial SARS-CoV-2 infection for the unvaccinated strata within the reinfection cohort.
[f]Exposure to individuals infected with SARS-CoV-2. [g]Reported personal protective equipment
(PPE) adherence at work was defined as the percentage of time in which an individual uses the
PPE recommended by their employer when in direct contact with people.
(DOCX)

**S2 Table. Count and descriptive statistics of quantitative antibody results for reinfection
correlates of protection cohort from AZHEROES/RECOVER, stratified by vaccination sta-
tus.**
(DOCX)

**S3 Table. Results from conditional logistic regression model with RBD and S2 AUC values
for first-time post-vaccination infection with Omicron following third COVID-19 origin
strain WA-1 monovalent mRNA vaccine dose correlates of protection cohort from
AZHEROES/RECOVER (n = 1226).** Abbreviations: OR: odds ratio; CI: confidence interval.
Odds ratio represents odds of being a case for each standard deviation increase in AUC. Cases
were defined as individuals who became infected with Omicron after receiving three origin
strain WA-1 monovalent COVID-19 vaccine doses and no prior infections. Cases and controls
were matched on number of days between blood draw and third vaccine dose, and study site.
*Statistically significant at alpha = 0.05.
(DOCX)

**S4 Table. Results from conditional logistic regression model with RBD and S2 AUC values
for the reinfection correlates of protection cohort from HEROES/RECOVER.** Abbrevia-
tions: OR: odds ratio; CI: confidence interval. Odds ratio represents odds of being a case for
each standard deviation increase in AUC. [a]Cases were defined as individuals who became rein-
fected with SARS-CoV-2 while unvaccinated. Both cases and controls were unvaccinated at
the time of their blood draw. [b]Cases were defined as individuals who became reinfected with
SARS-CoV-2. Both cases and controls were unvaccinated at time of initial infection, then
received 2 doses of an origin strain WA-1 monovalent COVID-19 vaccine prior to any poten-
tial reinfection. Blood draw for both cases and controls occurred after the 2nd dose. [c]Cases
were defined as individuals who became reinfected with SARS-CoV-2. Both cases and controls
were unvaccinated at time of initial infection, then received 3 doses of an origin strain WA-1
monovalent COVID-19 vaccine prior to any potential reinfection. Blood draw for both cases
and controls occurred after the 3rd dose. [d]At least one chronic condition versus no chronic
conditions. [e]Above cohort mean versus below cohort mean. *Statistically significant at
alpha = 0.05.
(DOCX)

**S5 Table. Unadjusted and adjusted odds ratios (ORs) generated by conditional logistic
regression models for a nested sample of individuals with first-time post-vaccination
infection with Omicron following third COVID-19 origin strain WA-1 monovalent
mRNA vaccine dose and matched controls from the HEROES/RECOVER prospective
cohort of frontline workers (n = 1226).** Abbreviations: OR: odds ratio; CI: confidence inter-
val. Odds ratio represents odds of being a case for each 3-fold increase in end titer. Cases were
defined as individuals who became infected with Omicron after receiving three COVID-19
vaccine doses. Cases and controls were matched on number of days between blood draw and
third vaccine dose, and study site. [a]At least one chronic condition versus no chronic condi-
tions. [b]Above cohort mean versus below cohort mean.*Statistically significant at alpha = 0.05.
(DOCX)

**S6 Table. Unadjusted and adjusted odds ratios (ORs) generated by conditional logistic regression models for a nested sample of individuals with reinfection and matched controls, stratified by vaccination status, from the HEROES/RECOVER prospective cohort of frontline workers (n = 740).** Abbreviations: OR: odds ratio; CI: confidence interval. Odds ratio represents odds of being a case for each 3-fold increase in end titer. [a]Cases were defined as individuals who became reinfected with SARS-CoV-2 while unvaccinated. Both cases and controls were unvaccinated at the time of their blood draw. [b]Cases were defined as individuals who became reinfected with SARS-CoV-2. Both cases and controls were unvaccinated at time of initial infection, then received 2 doses of an origin strain WA-1 monovalent COVID-19 vaccine prior to any potential reinfection. Blood draw for both cases and controls occurred after the 2nd dose. [c]Cases were defined as individuals who became reinfected with SARS-CoV-2. Both cases and controls were unvaccinated at time of initial infection, then received 3 doses of an origin strain WA-1 monovalent COVID-19 vaccine prior to any potential reinfection. Blood draw for both cases and controls occurred after the 3rd dose. [d]At least one chronic condition versus no chronic conditions. [e]Above cohort mean versus below cohort mean. *Statistically significant at alpha = 0.05.
(DOCX)

**S1 Fig.** Comparison of RBD (panel A) and S2 (panel B) AUC values between cases and controls in first-time post-vaccination infection with Omicron nested case-control cohort, with difference in means evaluated by paired t-test.
(TIFF)

**S2 Fig.** Comparison of mean RBD (panel A) and S2 (panel B) AUC values between cases and controls in reinfection Omicron infection nested case-control cohort, stratified by number of origin strain WA-1 monovalent mRNA COVID-19 vaccine doses.
(TIFF)

**S3 Fig. Flowchart for determining qualitative antibody result using RBD and S2 optical density values.**
(TIFF)

**S4 Fig. Flowchart for determining pre-enrollment infections in the HEROES-RECOVER cohort.**
(TIFF)

## Acknowledgments

CDC: Eduardo Azziz-Baumgartner, Melissa L. Arvay, William Brannen, Stephanie Bialek, Allison Ciesla, Alicia M. Fry, Aron Hall, Adam MacNeil, Clifford McDonald, Sue Reynolds, Robert Slaughter, Matthew J. Stuckey, Rose Wang, Ryan Wiegand; Abt Associates: Steve Pickett, Rekha Balachandran, Kim Groover, Deanna Fleary, Peenaz Mistry; BSWH: Kayan Dunnigan, Nicole Calhoun, Leah Odame-Bamfo, Clare Mathenge, Michael E. Smith, Kempapura Murthy, Tnelda Zunie, Eric Hoffman, Martha Zayed, Ashley Graves, Joel Blais, Jason Ettlinger, Sharla Russell, Natalie Settele, Tiya Searcy, Rupande Patel, Elisa Priest, Jennifer Thomas, Muralidhar Jatla, Madhava Beeram, Javed Butler, Alejandro Arroliga; KPNW: Holly Groom, Yolanda Prado, Daniel Sapp, Mi Lee, Chris Eddy, Matt Hornbrook, Donna Eubanks, Danielle Millay, Dorothy Kurdyla, Kristin Bialobok, Ambrosia Bass, Kristi Bays, Kimberly Berame, Cathleen Bourdoin, Rashyra Brent, Carlea Buslach, Lantoria Davis, Stephen Fortmann, Jennifer Gluth, Kenni Graham, Tarika Holness, Kelley Jewell, Enedina Luis, Abreeanah Magdaleno, DeShaun Martin, Joyce Smith-McGee, Martha Perley, Sam Peterson, Aaron Peipert, Krystil Phillips,

Joanna Price, Ana Reyes, Sperry Robinson, Katrina Schell, Emily Schield, Natosha Shirley, Anna Shivinsky, Valencia Smith, Britta Torgrimson-Ojerio, Brooke Wainwright, Shawn Westaway; Marshfield Clinic Research Laboratory: Saydee Benz, Adam Bissonnette, Krystal Boese, Emily Botten, Jarod Boyer, Michaela Braun, Julianne Carlson, Caleb Cravillion, Amber Donnerbauer, Tim Dziedzic, Joe Eddy, Heather Edgren, Alex Ermeling, Kelsey Ewert, Connie Fehrenbach, Rachel Fernandez, Wayne Frome, Sherri Guzinski, Mitch Hertel, Garrett Heuer, Erin Higdon, Cressa Huotari, Lynn Ivacic, Lee Jepsen, Steve Kaiser, Bailey Keffer, Tammy Koepel, Sarah Kohn, Alaura Lemieux, Carrie Marcis, Megan Maronde, Isaac McCready, Nidhi Mehta, Dan Miesbauer, Collin Nikolai, Brooke Olson, Jeremy Olstadt, Lisa Ott, Cory Pike, Nicole Price, Chris Reardon, Alex Slenczka, Elisha Stefanski, Lydia Sterzinger, Kendra Stoltz, Melissa Strupp, Lyndsay Watkins, Roxann Weigel, Ben Zimmerman; University of Miami: Damena Gallimore-Wilson, Roger Noriega, Cynthia Beaver, Alexandra Cruz, Annabel Reyes, Brigitte Madan, Addison Testoff, John Jones; St. Luke's: Jessica Lundgren, Karley Respet, Angela Hunt, Jennifer Viergutz, Daniel Stafki, Mary Robinson, Elizabeth Kaplan, Jill Dolezilek, Leiah Hoffman, Tyna O'Connor, Abbigail Hagen, Catherine Diluzio, Samantha Kendrick, Marilyn J. Odean; University of Arizona: Ariyah Armstrong, Nora Baccam, Zoe Baccam, Tatum Butcher, Shelby Capell, Andrea Carmona, Karysa Carson, Alissa Coleman, Hannah Cowling, Carly Deal, Kiara Earley, Sophie Evans, Julia Fisher, Ashlyn Flangos, Joe K. Gerald, Lynn Gerald, Anna Giudici, Erika Goebert, Taylor Graham, Sofia Grijalva, Hanna Hanson, Olivia Healy, Chloe Hendrix, Katherine Herder, Adrianna Hernandez, Raven Hilyard, Rezwana Islam, Krystal S. Jovel, Caroline Klinck, Karl Krupp, Karla Ledezma, Sally Littau, Amelia Lobos, Ashley Lowe, Jeremy Makar, Natalya Mayhew, Kristisha Mevises, Flavia Nakayima Miiro, Cierra Morris, Sarah Murray, Janko Nikolich-Žugich, Assumpta Nsengiyunva, Kennedy Obrien, Mya Pena, Riley Perlman, Celia Pikowski, Ferris A. Ramadan, Patrick Rivers, Jen Scott, Priyanka Sharma, Alison Slocum, Saskia Smidt, Lili Steffen, Jayla Sowell, Danielle Stea, Xiaoxiao Sun, Nicholas Tang, Gianna Taylor, Ta'Nya Tomas, Heena Timsina, Italia Trejo, April Yingst; University of Utah: Rachel T. Brown, Matthew Bruner, Riley Campbell, Aniysah Colston, Brianna Cottam, Chapman Cox, Kendal Chatard, Ben Cragun, Daniel Dawson, Anika Dsouza, Emilee Eden, Amanda Flanagan, Colton Fox, Adriele Fugal, Nikki Gallacher, Michelle Gillette, Joshua Griffin, Christian Guzman, Isaac Hansen, Addie Hunsaker, Iman Ibrahim, Nada Jabbouri, Junny Jeong, Ryder Jordin, Tori Joy, Michael Langston, Alexis Lowe, Katie Luong, Aurianna Martin, Gretchen Maughan, Jinyi Mao, Jeanmarie Mayer, Katy McCone, Renee McEntire, Jacob McKell, Tiana Miller, Max Minoughan, Naveen Naveed, Jessica Olsen, Christina Pick, Timina Powaukee, Jenna Praggastis, Lily Prentice, Seon Reed, Amber Rhodes, Griffin Rost, Camie Schaefer, Ashmita Shanthakumar, Madeleine Smith, Joseph Stanford, Grace Stewart, Pasha Stinson, Trevor Stubbs, Marcus Stucki, Madison Tallman, Sydney Taylor, Kathy Tran, Fiona Tsang, Irene Tucker, Maya Wheeler, Megan Wilson, Jesse Williams, Derrick Wong, Jenna Vo, Hannah Whiting, Iris Yang, Karly Anderson, Ian Arlington, Arlyne Arteaga, Braydon Black, Brock Bourdelle.

Special thanks to the firefighters, healthcare workers, and frontline and essential workers who participated in this cohort study.

## Author Contributions

**Data curation:** James Hollister, Ryan Sprissler.

**Formal analysis:** James Hollister, Cynthia Porter, Lauren Grant, Young M. Yoo.

**Funding acquisition:** Jefferey L. Burgess.

**Methodology:** James Hollister, Jennifer L. Uhrlaub, Lauren Grant, Katherine D. Ellingson.

**Project administration:** Amadea Britton.

**Resources:** Ryan Sprissler, Shawn C. Beitel.

**Supervision:** Ashley Fowlkes, Amadea Britton, Lauren E. W. Olsho, Manjusha Gaglani, Allison L. Naleway, Lisa Gwynn, Alberto Caban-Martinez, Harmony L. Tyner, Andrew L. Philips, Sarang Yoon, Karen Lutrick, Jefferey L. Burgess, Katherine D. Ellingson.

**Writing – original draft:** James Hollister, Cynthia Porter, Ryan Sprissler, Shawn C. Beitel, James K. Romine, Karen Lutrick, Jefferey L. Burgess, Katherine D. Ellingson.

**Writing – review & editing:** James Hollister, Cynthia Porter, Shawn C. Beitel, James K. Romine, Jennifer L. Uhrlaub, Lauren Grant, Young M. Yoo, Ashley Fowlkes, Amadea Britton, Lauren E. W. Olsho, Gabriella Newes-Adeyi, Sammantha Fuller, Pearl Q. Zheng, Manjusha Gaglani, Spencer Rose, Kayan Dunnigan, Allison L. Naleway, Lisa Gwynn, Alberto Caban-Martinez, Natasha Schaefer Solle, Harmony L. Tyner, Andrew L. Philips, Kurt T. Hegmann, Sarang Yoon, Karen Lutrick, Jefferey L. Burgess, Katherine D. Ellingson.

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
