## [Decision Letter · Decision Letter 0]

21 Aug 2024

PONE-D-24-25908Risk reduction in SARS-CoV-2 infection and reinfection conferred by humoral antibody levels among essential workers during Omicron predominancePLOS ONE

Dear Dr. Hollister,

Thank you for submitting your manuscript to PLOS ONE. After careful consideration, we feel that it has merit but does not fully meet PLOS ONE’s publication criteria as it currently stands. Therefore, we invite you to submit a revised version of the manuscript that addresses the points raised during the review process.

Statistical software is not properly referenced. Include manufacture, city/state and country in the statistical software used.

We look forward to receiving your revised manuscript.

Kind regards,

Maemu Petronella Gededzha, Ph.D

Academic Editor

PLOS ONE

Journal Requirements:

2. Thank you for stating the following in the Competing Interests section: "I have read the journal's policy and the authors of this manuscript have the following competing interests: RS reports a relationship with American Council of Life Insurers that includes: speaking and lecture fees. RS reports a relationship with California legal case Ebers v. Castle Park that includes: consulting or advisory. RS reports a relationship with Geneticure, Inc. that includes: equity or stocks. RS reports a relationship with Beckman Coulter that includes: speaking and lecture fees. RS reports a relationship with Shay Emma Hammer Research Foundation that includes: board membership. RS has patent issued to Arizona Board of Regents on Behalf of the University of Arizona. MG reports a relationship with Infectious Diseases and Immunization Committee, Texas Pediatric Society, Texas Chapter of the American Academy of Pediatrics that includes: board membership."

Reviewers' comments:

Reviewer's Responses to Questions

**Comments to the Author**

1. Is the manuscript technically sound, and do the data support the conclusions?

Reviewer #1: Yes

Reviewer #2: Yes

2. Has the statistical analysis been performed appropriately and rigorously? 

Reviewer #1: Yes

Reviewer #2: Yes

3. Have the authors made all data underlying the findings in their manuscript fully available?

Reviewer #1: Yes

Reviewer #2: No

4. Is the manuscript presented in an intelligible fashion and written in standard English?

Reviewer #1: Yes

Reviewer #2: Yes

5. Review Comments to the Author

Reviewer #1: The study examined the protection conferred by antibody levels against SARS-CoV-2 in frontline workers. Higher receptor binding domain (RBD) antibody titers were linked to a reduced risk of first-time post-vaccination infection and reinfection, especially during the Omicron variant's predominance. Vaccinated individuals with higher antibody levels had a significantly lower risk of reinfection compared to unvaccinated ones. Immunity was assessed using the SARS-CoV-2 Washington-1 spike protein but the current variants and subvariants may escape immunity generated with the ancestral strain which was rightfully acknowledged as a limitation. The paper contributes to understanding the SARS-COV2 correlates of protection.

Minor comments

In line 59-60 ‘first-time post-vaccination and reinfection study samples, most were female (67%, 57%), non -Hispanic (82%, 68%), and without chronic conditions (65%, 65%)’ verify the percentages. Do the same for line 249-250

In line 308-309 ‘the 1:60 dilution, estimated risk reduction was 50% at RBD and S2 end titers between 1:540 and 1:1620 (Fig 2)’. Refer to table 2 instead of figure 2

Questions for Clarification:

Was recurrent SARS-CoV-2 detected in any of the participants? Given the high exposure levels of the participants, one may expect reinfections to occur from the time blood was drawn.

Was the Omicron variant detected the same, or were various Omicron sub-variants detected in the participants? Provide details on the sequencing results.

Reviewer #2: The manuscript set out to report on the risk reduction and reinfection outcomes following vaccination and natural infection. The manuscript had a logical flow in information and have shown a reduction of 21% infection and 58% reinfection reduction as a result of the study vaccines used. It was not clear however in the discussion as to which vaccines were used in the comparison studies that they reported on. As there some differences in vaccine type as well as manufacturer with regards to immunological response.

6. PLOS authors have the option to publish the peer review history of their article (what does this mean?). If published, this will include your full peer review and any attached files.

Reviewer #1: No

Reviewer #2: **Yes: **Omphile E Simani

---

## [Author Response · Author response to Decision Letter 0]

29 Oct 2024

Reviewer #1:

(1) In line 59-60 ‘first-time post-vaccination and reinfection study samples, most were female (67%, 57%), non-Hispanic (82%, 68%), and without chronic conditions (65%, 65%)’ verify the percentages. Do the same for line 249-250.

Response: We have confirmed that these percentages are correct. These percentages were calculated by combining cases and controls to save on word count in the abstract, as well as to provide a general summary of our cohort while providing stratified information in table 1.

(2) In line 308-309 ‘the 1:60 dilution, estimated risk reduction was 50% at RBD and S2 end titers between 1:540 and 1:1620 (Fig 2)’. Refer to table 2 instead of figure 2.

Response: Thank you for catching this. We actually meant to refer to figure 3 for this text, as this figure shows where the estimated risk reduction reaches 50% in this population. We have updated this in the text. In addition, we incorrectly refer to figure 3 rather than figure 4 in the reinfection results section. We have updated this in the text as well.

(3) Was recurrent SARS-CoV-2 detected in any of the participants? Given the high exposure levels of the participants, one may expect reinfections to occur from the time blood was drawn.

Response: For the first-time Omicron infection cohort, we did not assess reinfections in this specific manuscript as it fell outside of the study period. However, we did observe Omicron reinfections in the general AZHEROES/RECOVER cohort, and understanding reinfections was a point of emphasis for the larger study. We invite the reviewer to review a manuscript that we previously published that assessed risk factors for Omicron reinfection among frontline workers in the AZHEROES/RECOVER cohort: https://dx.doi.org/10.3201/eid2903.221314.

(4) Was the Omicron variant detected the same, or were various Omicron sub-variants detected in the participants? Provide details on the sequencing results.

Response: For our study set, approximately half of the nasal swabs among cases were sequenced. Among the samples that were sequenced, we identified a total of 34 unique Pango lineages for the Omicron variant, with the 5 most-common being BA.1.1, BA.1, BA.2.12.1, BA.2, and BA.5.2.1. However, because not all nasal swabs were sequenced, and because we only imputed variant of infection for those that were not sequenced (ancestral, Delta, or Omicron), we did not conduct any secondary analyses that divided our cases by lineage. 

Reviewer #2:

(1) It was not clear however in the discussion as to which vaccines were used in the comparison studies that they reported on. As there some differences in vaccine type as well as manufacturer with regards to immunological response.

Response: Thank you for your review. There are differences in immunological response by vaccination manufacturer, and we agree that this is an important detail to consider when comparing results. We have added vaccine information in the second paragraph of our discussion to help contextualize the studies we are citing to compare our results.

---

## [Decision Letter · Decision Letter 1]

16 Dec 2024

Risk reduction in SARS-CoV-2 infection and reinfection conferred by humoral antibody levels among essential workers during Omicron predominance

PONE-D-24-25908R1

Dear Dr. Hollister,

We’re pleased to inform you that your manuscript has been judged scientifically suitable for publication and will be formally accepted for publication once it meets all outstanding technical requirements.

Kind regards,

Maemu Petronella Gededzha, Ph.D

Academic Editor

PLOS ONE

Additional Editor Comments (optional):

The comments have been addressed to the satisfaction of the reviewers and the editor

Reviewers' comments:

Reviewer's Responses to Questions

**Comments to the Author**

1. If the authors have adequately addressed your comments raised in a previous round of review and you feel that this manuscript is now acceptable for publication, you may indicate that here to bypass the “Comments to the Author” section, enter your conflict of interest statement in the “Confidential to Editor” section, and submit your "Accept" recommendation.

Reviewer #1: All comments have been addressed

2. Is the manuscript technically sound, and do the data support the conclusions?

Reviewer #1: Yes

3. Has the statistical analysis been performed appropriately and rigorously? 

Reviewer #1: Yes

4. Have the authors made all data underlying the findings in their manuscript fully available?

Reviewer #1: (No Response)

5. Is the manuscript presented in an intelligible fashion and written in standard English?

Reviewer #1: Yes

6. Review Comments to the Author

Reviewer #1: The authors have made all reviewers recommendations and the paper is acceptable for publication, no further comments

7. PLOS authors have the option to publish the peer review history of their article (what does this mean?). If published, this will include your full peer review and any attached files.

Reviewer #1: No

---

## [Editor Report · Acceptance letter]

18 Dec 2024

PONE-D-24-25908R1 

PLOS ONE

Dear Dr. Hollister, 

I'm pleased to inform you that your manuscript has been deemed suitable for publication in PLOS ONE. Congratulations! Your manuscript is now being handed over to our production team.

Kind regards, 

on behalf of

Dr. Maemu Petronella Gededzha 

Academic Editor

PLOS ONE